# Towards a unified molecular mechanism for ligand-dependent activation of NR4A-RXR heterodimers

Xiaoyu Yu[1,2], Yuanjun He[3], Thedore M Kamenecka[3], Douglas J Kojetin[1,2,3,4,5,6]*

[1]Department of Biochemistry, Vanderbilt University, Nashville, United States; [2]Department of Integrative Structural and Computational Biology, Scripps Research and The Herbert Wertheim UF Scripps Institute for Biomedical Innovation and Technology, Jupiter, United States; [3]Department of Molecular Medicine, Scripps Research and The Herbert Wertheim UF Scripps Institute for Biomedical Innovation and Technology, Jupiter, United States; [4]Center for Structural Biology, Vanderbilt University, Nashville, United States; [5]Vanderbilt Institute of Chemical Biology, Vanderbilt University, Nashville, United States; [6]Center for Applied AI in Protein Dynamics, Vanderbilt University, Nashville, United States

## eLife Assessment

This **important** study investigated whether the nuclear receptor Nur77 is regulated by a non-canonical mechanism of ligand-induced disruption of its interaction with RXRg, similar to the family member Nurr1. The overall evidence is **compelling**. This manuscript will be of interest to scientists focusing on mechanisms of transcriptional regulation.

*For correspondence:
douglas.kojetin@vanderbilt.edu

Competing interest: The authors declare that no competing interests exist.

**Abstract** A subset of nuclear receptors (NRs) function as permissive heterodimers with retinoid X receptor (RXR), defined by transcriptional activation in response to RXR agonist ligands. Permissive NR-RXR activation is generally understood to operate through a classical pharmacological mechanism in which RXR agonist binding enhances coactivator recruitment to the heterodimer. However, we previously demonstrated that transcriptional activation of permissive Nurr1-RXRα (NR4A2-NR2B1) heterodimers by an RXR ligand set, which included pharmacological RXR agonists and selective Nurr1-RXRα agonists that function as antagonists of RXRα homodimers, is explained by a non-classical activation mechanism involving ligand-binding domain (LBD) heterodimer dissociation (Yu et al., 2023). Here, we extend mechanistic ligand profiling of the same RXR ligand set to the evolutionarily related Nur77-RXRγ (NR4A1-NR2B3) heterodimer. Biochemical and NMR protein-protein interaction profiling, together with cellular transcription studies, indicate that activation of Nur77-RXRγ transcription by the RXR ligand set, which lacks selective Nur77-RXRγ agonists, is consistent with contributions from both classical pharmacological activation and LBD heterodimer dissociation. However, reanalysis of our previously published data for Nurr1-RXRα revealed that inclusion of selective Nurr1-RXRα agonists was essential for elucidating the LBD heterodimer dissociation mechanism. Together, our findings highlight the importance of using a more functionally diverse RXR ligand set to define the mechanism of Nur77-RXRγ activation and to further evaluate whether LBD heterodimer dissociation represents a shared activation mechanism among NR4A-RXR heterodimers relevant to neurodegenerative and inflammatory diseases.

## Introduction

The orphan nuclear receptor Nur77 (NR4A1) is implicated as a drug target in CNS and neurological disorders including stroke and Parkinson's disease (*Liu et al., 2021*; *Saucedo-Cardenas and Conneely, 1996*) and chronic inflammatory and autoimmune disorders (*Lith and de Vries, 2021*). Nur77 plays important roles in maintaining the health, development, and homeostasis of the brain by regulating autophagy (*Ping et al., 2021*), neuroinflammation (*Yan et al., 2020*), and dopamine turnover and transmission (*Lévesque and Rouillard, 2007*). Nur77-deficient mice exhibit higher levels of dopamine metabolite DOPAC and reduced levels of dopamine alongside increased spontaneous locomotor activity (*Gilbert et al., 2006*; *Mount et al., 2013*). Nur77 is protective against Parkinsonian symptoms induced by neurotoxin 1-methyl-4-phenyl-1,2,3,6-tetrahydropyridine (MPTP) treatment (*Mount et al., 2013*), emphasizing the relevance of Nur77 to neurodegenerative disease. Nur77 activates the NLRP3 inflammasome through a non-transcriptional regulation mechanism through direct binding to NLRP3 (*Zhu et al., 2023*) and anti-tumor immunity via transcriptional regulation (*Liebmann et al., 2018*; *Mandula et al., 2024*). Nur77 also regulates oxidative metabolism in skeletal muscle tissues and cells (*Chao et al., 2012*; *Chao et al., 2007*) and plays an important role in thymic negative selection in T-cells (*Liebmann et al., 2018*; *Lith et al., 2020*).

The molecular basis for the druggability of NRs is primarily encoded within a specific region, the ligand-binding domain (LBD), which has evolved to bind endogenous cellular metabolites and, in turn, is the target for approximately 15% of FDA-approved drugs (*Santos et al., 2017*). The orthosteric ligand-binding pocket is a solvent-occluded hydrophobic region located in the core of the LBD. Agonist binding to the orthosteric pocket stabilizes the NR LBD in a conformation that enables high-affinity binding and recruitment of transcriptional coactivator proteins to increase transcription and gene expression (*Kojetin and Burris, 2013*)—the classical pharmacological activation mechanism—but can also impact nuclear import/export and subcellular localization, post-translational modification, interactions with other proteins, among other activities. However, crystal structures of the LBDs of Nur77 and an evolutionarily related NR4A subfamily member, Nurr1 (NR4A2), show that their orthosteric ligand-binding pockets are filled with bulky hydrophobic residues (*Flaig et al., 2005*; *Wang et al., 2003*), which may prevent ligand access. Progress has been made to identify putative Nur77 and Nurr1 endogenous cellular metabolites (*Bruning et al., 2019*; *de Vera et al., 2016*; *de Vera et al., 2019*; *Rajan et al., 2020*; *Vinayavekhin and Saghatelian, 2011*) and synthetic ligands (*Jang et al., 2021*; *Munoz-Tello et al., 2020*; *Safe et al., 2021*; *Willems and Merk, 2022*) that bind to their LBDs and regulate their activities. However, since the discovery of Nurr1 and Nur77 in the late 1980s and early 1990s (*Hazel et al., 1988*; *Law et al., 1992*), small molecule targeting the NR4A receptors remains challenging compared to other prototypical NRs (*Burris et al., 2013*), leading the field to seek out alternative mechanisms to regulate NR4A activities.

Nurr1 and Nur77 can regulate transcription as monomers or as a permissive heterodimer with retinoic X receptors RXRα (NR2B1) and RXRγ (NR2B3), respectively, which form heterodimers with approximately one-third of the NR superfamily (*Lefebvre et al., 2010*; *Lévesque and Rouillard, 2007*). Thus, one such alternative mechanism to target NR4A transcription that has been explored in the field is through ligands binding to their RXR heterodimer partner (*Asvos et al., 2025*; *Loppi et al., 2018*; *McFarland et al., 2013*; *Spathis et al., 2017*; *Wallen-Mackenzie et al., 2003*; *Wang et al., 2016*). NR-RXR heterodimers are classified as permissive or non-permissive depending on whether they can be activated by RXR agonist binding to RXR (permissive) or not (non-permissive). Pharmacological activation of NRs is understood to occur via a classical model where agonist binding stabilizes an active LBD conformation that promotes recruitment of transcriptional coactivator proteins, ultimately resulting in increased gene expression. However, we recently showed that transcriptional activation of Nurr1-RXRα heterodimers by pharmacological RXR agonists or selective Nurr1-RXRα agonists, which function as antagonists of RXRα homodimers, occurs through a non-classical model: LBD heterodimer protein-protein interaction (PPI) inhibition (*Yu et al., 2023*).

Building on our published Nurr1-RXRα study, we hypothesized that Nur77 activity could similarly be modulated by ligand binding to its heterodimer partner, RXRγ. Here, we tested the same RXR ligand set used in our Nurr1-RXRα study using the same comparative Nur77-RXRγ structure-function analyses. Together, our published and new findings show that RXRγ and RXRα function as repressive heterodimer partners for Nur77 and Nurr1, respectively. Ligand profiling studies also indicate that RXR ligands may influence Nur77-RXRγ heterodimer transcription through LBD heterodimer

dissociation. However, nuanced observations in the Nur77-RXRγ ligand profiling data—and the lack of selective Nur77-RXRγ agonists that, similar to selective Nurr1-RXRα agonists, would function as antagonists of RXR homodimers—limit the ability to determine the extent to which Nur77-RXRγ activation reflects contributions from LBD heterodimer dissociation, classical pharmacological activation, or both mechanisms.

## Results

### RXRγ LBD is required for repression of Nur77-mediated transcription

In our recent study (*Yu et al., 2023*), we confirmed published observations (*Aarnisalo et al., 2002*; *Forman et al., 1995*) that RXRα represses Nurr1-mediated transcription on a monomeric Nurr1-binding DNA response element called NBRE, which is present in the promoter regions of Nurr1-regulated genes in the dopamine biosynthesis pathway. Using RXRα domain truncation analysis, we previously found that RXRα-induced repression occurs via the RXRα LBD interaction with Nurr1, consistent with the well-understood mechanism that LBD-LBD interactions constitute the primary basis for NR homodimerization and heterodimerization. We also showed that RXR ligands that activate

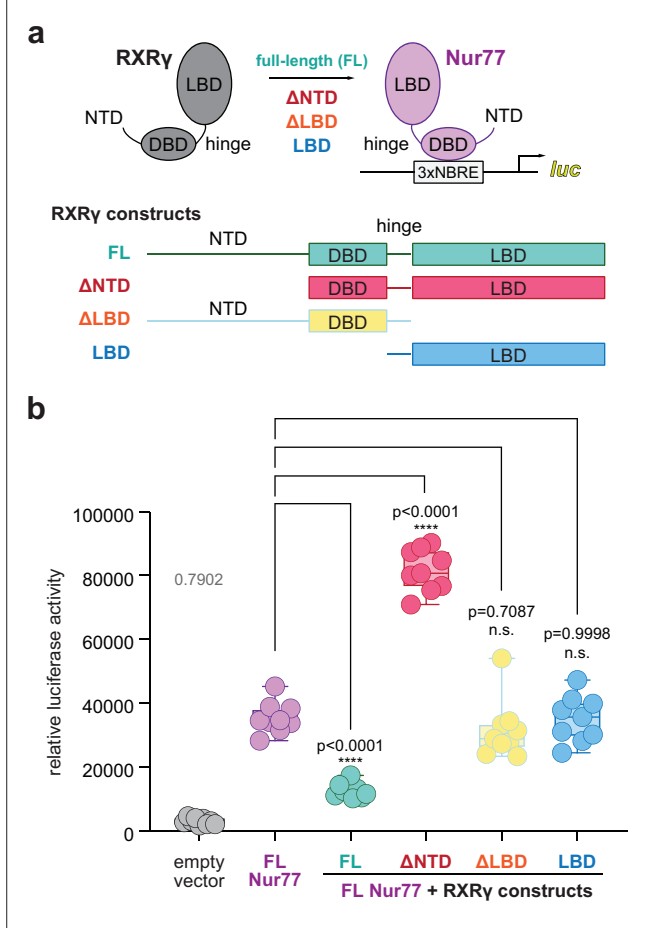

**Figure 1.** Contribution of RXRγ domains on repressing Nur77-mediated transcription. (**a**) General scheme of the cellular transcriptional reporter assay. (**b**) 3xNBRE-luciferase assay performed in SK-N-BE(2)-C cells. Data are normalized to empty vector control (n=9 replicates), shown as a box and whiskers plot with boundaries of the box representing the 25th percentile and the 75th percentile, and representative of three independent experiments. Statistical testing was performed and p-values were calculated using the Brown-Forsythe and Welch multiple comparisons test of the FL Nur77+RXRγ constructs conditions relative to the FL Nur77 control condition. See *Figure 1—source data 1* for data plotted.

The online version of this article includes the following source data for figure 1:

**Source data 1.** Data underlying the plot in *Figure 1b*.

transcription of Nurr1-RXRα heterodimers function via a mechanism involving LBD-LBD heterodimer PPI inhibition, freeing a transcriptionally active Nurr1 monomer from the repressive Nurr1-RXRα heterodimer. This was in contrast to the prevailing mechanism in the field for RXR ligand-induced activation of permissive heterodimers where, for example, RXRα contributes to activation of PPARγ-mediated transcription (*Kliewer et al., 1992*), and agonist binding to either or both NRs synergistically activates PPARγ-RXRα heterodimer-mediated transcription via enhanced coactivator recruitment (*de Vera et al., 2017*; *DiRenzo et al., 1997*; *Kojetin et al., 2015*).

To test if RXRγ represses Nur77-mediated transcription through a similar mechanism with Nurr1-RXRα, we performed a transcriptional reporter assay where SK-N-BE(2) neuronal cells were transfected with a full-length Nur77 expression plasmid, with or without full-length or domain-truncation RXRγ expression plasmids (*Figure 1a*), along with a plasmid containing three copies of the monomeric NBRE DNA-binding response element sequence upstream of the luciferase gene (3xNBRE-luc). Similar to our Nurr1-RXRα findings (*Yu et al., 2023*), cotransfection of RXRγ (FL) repressed Nur77-mediated transcription, whereas an RXRγ construct lacking the LBD (ΔLBD) had no effect (*Figure 1b*). Cotransfection of an RXRγ construct lacking the N-terminal AF-1 domain (ΔNTD) activated Nur77-mediated transcription, whereas a construct lacking the NTD and DNA-binding domain containing the hinge region (RXRγ-LBD) had no effect.

These data, which indicate that both the RXRγ NTD and LBD are necessary for repression of Nur77-mediated transcription, are consistent with a model in which LBD-dependent PPIs contribute to RXR-mediated repression of Nur77-RXRγ and Nurr1-RXRα heterodimers. However, there are additional nuances that should be considered in the interpretation of these data. First, the RXRγ NTD is a principal domain that drives RXRγ phase separation in vitro and biomolecular condensates in cells (*Sołtys et al., 2025*; *Sołtys et al., 2021*; *Wang et al., 2025*). Therefore, perturbing RXRγ's ability to form cellular condensates via removal of its NTD may change how RXRγ interacts with Nur77, which also forms cellular condensates (*Chen et al., 2025*; *Chen et al., 2024*; *Peng et al., 2021*), or coregulator proteins that are recruited to condensates containing Nur77 and/or RXRγ. Furthermore, the NTD of nuclear receptors can interact with coactivator and corepressor proteins (*Dotzlaw et al., 2002*; *Hodgson et al., 2005*; *Simons et al., 2014*; *Wang and Simons, 2005*), including a study that showed the Nur77 NTD interacts with the FHL2 corepressor (*Kurakula et al., 2011*). Although we are not aware of published studies showing the RXRγ NTD interacts with corepressor proteins, in principle, truncation of the RXRγ NTD could show higher transcription via decreased corepressor interaction to Nur77-RXRγ(ΔNTD) heterodimers relative to wild-type Nur77-RXRγ heterodimers. Together, these and other limitations of the RXRγ domain truncation strategy—including potential differences in protein expression levels and truncation-dependent changes in subcellular localization, which we did not assess—limit a full mechanistic interpretation of these data. Despite these limitations, the truncation data support a conclusion that both the RXRγ NTD and LBD contribute to repression of Nur77-mediated transcription and thus point to a more complex domain-level mechanism of Nur77-RXRγ regulation than observed for Nurr1-RXRα.

## RXR ligand profiling of Nur77-RXRγ heterodimer activation vs. pharmacological RXRγ agonism

We next determined the influence of RXR ligands on Nur77-RXRγ transcriptional activity using the 3xNBRE luciferase reporter assay in human SK-N-BE(2) neuronal cells (*Figure 2a*). We used the same RXR ligand set (*Figure 2—figure supplement 1*) from our previous study on Nurr1-RXRα (*Yu et al., 2023*) composed of pharmacological RXR agonists (9-cis retinoic acid, Bexarotene, CD3254, IRX4204, LG100268) and antagonists (Danthron, HX531, PA452, Rhein, UVI3003), two compounds reported to selectively activate Nurr1-RXRα heterodimers (BRF110 and HX600), and one compound reported to selectively activate PPAR-RXR and RAR-RXR heterodimers (LG100754). Similar to our findings for Nurr1-RXRα, pharmacological RXR agonists activated Nur77-RXRγ transcription—however, one of the RXR agonists, IRX4204, activated Nur77-RXRγ but not Nurr1-RXRα (*Yu et al., 2023*), which may be explained by its 2.5-fold potency for RXRγ compared to RXRα; compound 2 in *Vuligonda et al., 2001*. Pharmacological RXR antagonists did not affect Nur77-RXRγ transcription, consistent with our Nurr1-RXRα findings. For two selective Nurr1-RXRα agonists, BRF110 and HX600, which function as RXRα homodimer antagonists, HX600 showed partial activation of Nur77-RXRγ whereas BRF110 did not activate, consistent with the original report of BRF110 as a selective agonist for Nurr1-RXRα (*Spathis et al., 2017*).

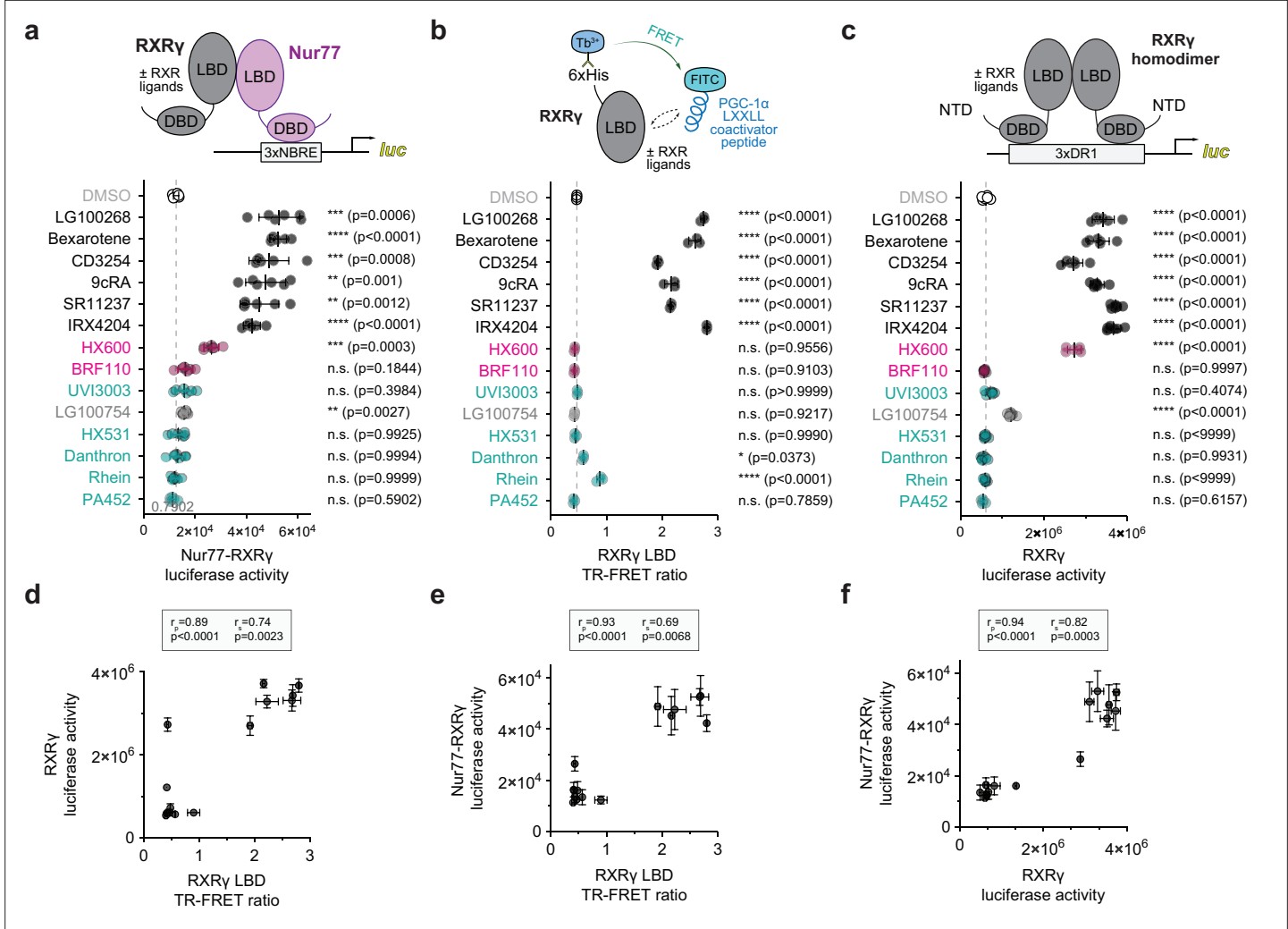

**Figure 2.** Ligand profiling for Nur77-RXRγ heterodimer activation and pharmacological RXRγ agonism. (**a**) General scheme and data from the Nur77-RXRγ/3xNBRE-luciferase cellular transcriptional reporter assay performed in SK-N-BE(2)-C cells treated with RXR ligand (1 µM) or DMSO (dotted line). Data are normalized to DMSO (n=6 replicates), represent the mean ± s.d., and representative of two independent experiments. See *Figure 2—source data 1* for data plotted. (**b**) General scheme and data from RXRγ LBD time-resolved fluorescence resonance energy transfer (TR-FRET) coactivator peptide interaction assay. TR-FRET ratio measured in the presence of DMSO (dotted line) or compound (2–4 µM). Data are normalized to DMSO control (n=3 replicates), represent the mean ± s.d., representative of two independent experiments. See *Figure 2—source data 2* for data plotted. (**c**) General scheme and data from the RXRγ/3xDR1-luciferase cellular transcriptional reporter assay performed in HEK293T cells treated with compound (1 µM) or DMSO control (dotted line). Data normalized to DMSO (n=6 replicates), represent the mean ± s.d., and representative of two independent experiments. See *Figure 2—source data 3* for data plotted. (**d,e,f**) Correlation plots of (**d**) RXRγ transcriptional reporter data vs. RXRγ LBD TR-FRET data, (**e**) Nur77-RXRγ cellular transcription data vs. RXRγ LBD TR-FRET data, and (**f**) Nur77-RXRγ cellular transcription data vs. RXRγ transcriptional reporter data with calculated Pearson ($r_p$) and Spearman ($r_s$) correlation coefficients. For all comparisons, statistical testing was performed, and p-values were calculated, using the Brown-Forsythe and Welch (**a,c**) or ordinary one-way ANOVA (**b**) tests for multiple comparisons with Dunnett corrections relative to DMSO control treated condition. Data and RXR ligand label text in (**a,b,c**) are colored according to RXR ligand activity as grouped in *Figure 2—figure supplement 1*.

The online version of this article includes the following source data and figure supplement(s) for figure 2:

**Source data 1.** Data underlying the plot in *Figure 2a*.

**Source data 2.** Data underlying the plot in *Figure 2b*.

**Source data 3.** Data underlying the plot in *Figure 2c*.

**Figure supplement 1.** RXR ligand set used in this study.

We next profiled how the RXR ligands affect transcriptional activation of RXRγ homodimers using biochemical and cellular assays similar to the assays we performed for RXRα (*Yu et al., 2023*). We used a time-resolved fluorescence resonance energy transfer (TR-FRET) biochemical assay to determine how the ligands influence the interaction between RXRγ LBD and a peptide derived from PGC1α (*Figure 2b*), a coactivator that regulates RXR-mediated transcription (*Delerive et al., 2002*). We also determined how the ligands influence transcription mediated by RXRγ homodimers (*Figure 2c*). Similar to our findings for RXRα, the RXR agonists enhanced PGC1α recruitment to the RXRγ LBD and increased RXRγ transcriptional activity, whereas antagonists had no effect. BRF110 did not affect RXRγ transcriptional activity; however, HX600 activated RXRγ-mediated transcription, consistent with a previous study showing that HX600 displays high activity for RXRγ but moderate activity for RXRα (*Umemiya et al., 1997*). However, HX600 did not promote recruitment of the PGC1α coactivator peptide in the TR-FRET assay, indicating that the activation of RXRγ transcription by HX600 may occur through either a coactivator recruitment-independent mechanism or through recruiting other coactivators among other possibilities, which would need further study.

In our previous study on Nurr1-RXRα, using Pearson (linear) and Spearman (monotonic relationship) analysis we found that Nurr1-RXRα mediated transcription by the RXR ligand set is not correlated to features of pharmacological RXRα agonism—that is, RXRα homodimer transcription or RXRα LBD coactivator peptide recruitment in a TR-FRET biochemical assay—but instead show moderate-to-strong correlations to weakening Nurr1-RXRα LBD interaction and binding affinity in the NMR and ITC data, respectively (*Yu et al., 2023*). In contrast, a strong correlation was observed between RXRα homodimer transcription and RXRα LBD TR-FRET coactivator peptide recruitment, consistent with the classical pharmacological activation mechanism. Similarly, we observed a strong correlation is observed between RXRγ transcription and RXRγ LBD TR-FRET coactivator peptide recruitment mediated by the RXR ligand set (*Figure 2d*), indicating RXRα and RXRγ share a similar classical pharmacological activation mechanism. However, in contrast to what we reported for Nurr1-RXRα, we found that Nur77-RXRγ transcription imparted by the RXR ligand set is strongly correlated to coactivator peptide recruitment to the RXRγ LBD in a TR-FRET biochemical assay (*Figure 2e*) and transcriptional activation of full-length RXRγ (*Figure 2f*). These data suggest that RXR ligands may regulate Nur77-RXRγ transcription through the classical pharmacological activation mechanism, which is distinct from Nurr1-RXRα, leading us to test if these RXR ligands can dissociate Nur77-RXRγ as they do to Nurr1-RXRα (*Yu et al., 2023*).

## RXR ligand profiling of Nur77-RXRγ heterodimer activation vs. LBD heterodimer dissociation

In our previous study, the RXR ligand-induced Nurr1-RXRα LBD heterodimer dissociation mechanism was apparent via three structural biology and biophysical methods (*Yu et al., 2023*). 2D [$^1$H,$^{15}$N]-TROSY-HSQC NMR structural footprinting showed that the addition of Nurr1-RXRα activating ligands to $^{15}$N-labeled Nurr1 LBD heterodimerized with RXRα LBD resulted in the appearance of a population of Nurr1 LBD monomer NMR peaks where monomer population occupancy strongly correlated to RXR ligand-induced Nurr1-RXRα transcriptional activation. We therefore performed NMR structural footprinting to determine the impact of the RXR ligand set on Nur77-RXRγ LBD heterodimer interaction.

We first performed multi-angle light scattering (MALS) at a protein concentration (180 μM) similar to our NMR samples (200–400 μM), which confirmed the Nur77 LBD and Nurr1 LBD are monomeric in solution, whereas RXRα and RXRγ LBDs are predominately homodimers with a minor homotetramer species (*Figure 3—figure supplement 1*) consistent with published studies on RXR (*Gampe et al., 2000*; *Kersten et al., 1997*). Because there are no published NMR chemical shift assignments for Nur77 LBD, we expressed and purified [$^2$H,$^{15}$N,$^{13}$C]-labeled Nur77 LBD, collected three-dimensional (3D) NMR backbone assignment experiments, and assigned ~85% of non-proline chemical shifts. NMR chemical shift footprinting analysis showed that the addition of RXRγ LBD to $^{15}$N-labeled Nur77 LBD revealed a transition in 2D [$^1$H,$^{15}$N]-TROSY-HSQC NMR data from a Nur77 LBD monomer species to a Nur77-RXRα LBD heterodimeric form (*Figure 3—figure supplement 2*). Addition of RXR ligands resulted in select chemical shift perturbations (CSPs) where a population of NMR peaks corresponding to both the Nur77 LBD monomer and the Nur77-RXRγ heterodimer were observed. This peak doubling phenomenon was apparent for several Nur77 residues with well-resolved NMR peaks, including G544 (*Figure 3a*) and G376 (*Figure 3—figure supplement 3*) located on helix 9 near

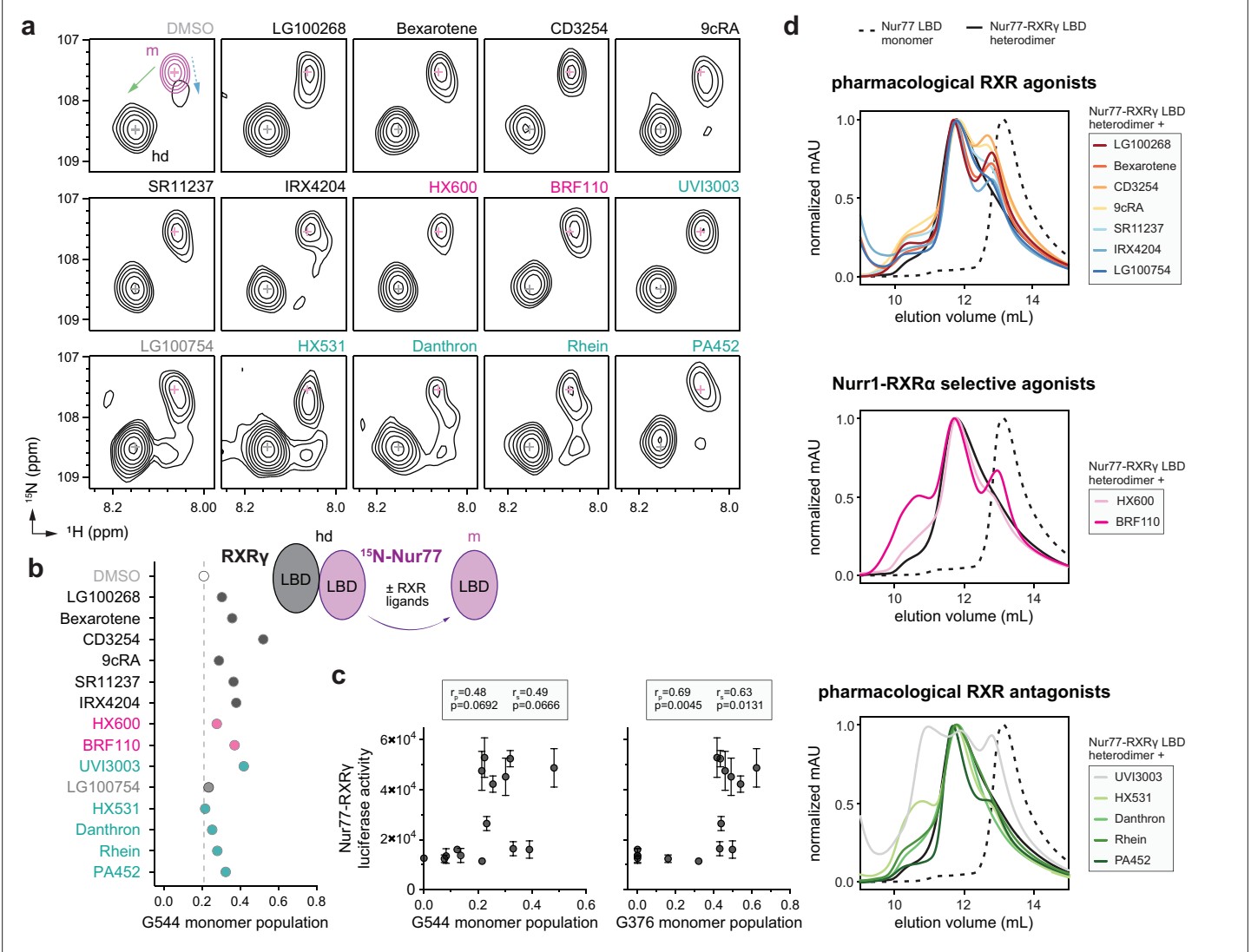

**Figure 3.** Ligand profiling for Nur77-RXRγ LBD heterodimer dissociation. (**a**) 2D [$^1$H,$^{15}$N]-TROSY HSQC data of $^{15}$N-labeled Nur77 LBD heterodimerized with unlabeled RXRγ LBD in the presence of RXR ligands focused on the NMR peak of G544. The upper left shows an overlay of two spectra corresponding to $^{15}$N-labeled Nur77 LBD monomer (200 μM; purple) and $^{15}$N-labeled Nur77 LBD + unlabeled RXRγ LBD heterodimer (1:2 molar ratio; black) to demonstrate the shift of the G544 peak between Nur77 LBD monomer (m) and heterodimer (hd) forms; solid green and dotted blue arrows denote the complex chemical shift perturbation pattern. RXR ligand label text is colored according to RXR ligand activity as grouped in **Figure 2— figure supplement 1**. (**b**) Peak intensity estimated ligand-dependent Nur77 LBD monomer populations from G544 and G376 in the 2D [$^1$H,$^{15}$N]-TROSY HSQC data (n=1). Data and RXR ligand label text are colored according to RXR ligand activity as grouped in **Figure 2—figure supplement 1**. See **Figure 3—source data 1** for data plotted. (**c**) Correlation plot of Nur77-RXRγ cellular transcription data vs. NMR estimated Nur77 LBD monomer populations for G544 and G376 with calculated Pearson ($r_p$) and Spearman ($r_s$) correlation coefficients; see **Figure 3—figure supplement 3** for G376 2D [$^1$H,$^{15}$N]-TROSY HSQC data. (**d**) Analytical size exclusion chromatography (SEC) analysis of Nur77-RXRγ LBD in the presence of RXR ligands (solid colored lines) relative to Nur77 LBD monomer (dotted black line) and Nur77-RXRγ LBD heterodimer (solid black line; n=1).

The online version of this article includes the following source data and figure supplement(s) for figure 3:

**Source data 1.** Data underlying the plot in **Figure 3b**.

**Figure supplement 1.** Multiangle light scattering (MALS) analysis of the ligand-binding domains (LBDs) of Nurr1, Nur77, RXRα, and RXRγ.

**Figure supplement 2.** Overlay of 2D [$^1$H,$^{15}$N]-TROSY HSQC data of $^{15}$N-labeled Nur77 LBD heterodimerized with unlabeled RXRγ LBD.

**Figure supplement 3.** Additional NMR ligand profiling for Nur77-RXRγ LBD heterodimer dissociation focused on Nur77 residue G376.

**Figure supplement 4.** AlphaFold3 structural model of the Nur77-RXRγ LBD heterodimer used to highlight the locations of G376 and G544 that were analyzed in the NMR analysis.

**Figure supplement 5.** Isothermal titration calorimetry (ITC) analysis of Nur77 LBD titrated into RXRγ LBD at the indicated temperatures.

*Figure 3 continued on next page*

*Figure 3 continued*

**Figure supplement 6.** Isothermal titration calorimetry (ITC) analysis of Nur77 LBD titrated into RXRγ LBD performed at 5 °C in the presence of the indicated RXR ligands or DMSO (vehicle control).

the heterodimerization surface and helix 1 distal from the heterodimerization surface, respectively (*Figure 3—figure supplement 4*).

In our previous study on Nurr1-RXRα, addition of RXR ligands resulted in the appearance of monomeric Nurr1 NMR peaks that were in slow exchange on the NMR time scale, where the monomeric and heterodimeric species were long-lived, allowing an estimation of the relative population of dissociated monomeric Nurr1 via peak integration analysis (*Yu et al., 2023*). However, NMR data of RXR ligands added to $^{15}$N-labeled Nur77 heterodimerized to RXRγ LBD are more complex, showing mixed features of slow exchange, intermediate-to-fast exchange, and multiple populations and peak lineshapes that include peaks shifted off the diagonal between Nur77 in the monomer and heterodimer states—and is also evident in the NMR spectrum of the heterodimer complex in the absence of ligand. At two molar equivalents of added RXRγ LBD, a shift to the heterodimer species (*Figure 3a*, green arrow) is evident with a small population in exchange with the Nur77 monomer present shifted in a different direction from the heterodimer peak (*Figure 3a*, blue arrow). These data suggest that RXRγ LBD binding to Nur77 LBD may occur through a binding exchange mechanism that includes an induced fit or a conformational change after binding component. Notwithstanding these challenges, we estimated the relative population of dissociated monomeric Nur77 LBD using measured NMR peak volumes before and after addition of RXR ligand (*Figure 3b*), which revealed that all the RXR ligands dissociated the Nur77-RXRγ LBD heterodimer to some degree although it is important to note that this analysis is confounded by the mix of exchange and peak lineshape properties—this system does not reveal isolated monomer vs. heterodimer populations as we observed for Nurr1-RXRα. Pearson (linear) and Spearman (monotonic relationship) analysis reveals a strong correlation between the RXR ligand-induced appearance of NMR-detected Nur77 monomer populations and Nur77-RXRγ transcription activation, which indicates RXR ligands activating Nur77-RXRγ transcription may also promote heterodimer dissociation (*Figure 3c*).

In our previous study on Nurr1-RXRα, isothermal titration calorimetry (ITC) studies showed a strong correlation between ITC-measured reduction in Nurr1-RXRα LBD heterodimerization affinity and transcriptional activation of Nurr1-RXRα for the RXR ligand set (*Yu et al., 2023*). We therefore used ITC to quantify how RXR ligands influence heterodimerization binding affinity between Nur77 LBD and RXRγ LBD. ITC experiments showed no measurable enthalpic component to the Nur77-RXRγ LBD interaction in the absence of ligand at 25 °C, a weak enthalpic component to binding at 15 °C, and a more notable enthalpic component at 5 °C (*Figure 3—figure supplement 5*). We could faithfully determine a binding affinity for the apo/ligand-free Nur77-RXRγ LBD heterodimer at 5 °C ($K_d$ = 9 μM); by comparison, ITC-determined Nurr1-RXRα LBD heterodimer affinity is ~4 μM at 25 °C (*Yu et al., 2023*). However, addition of select RXR ligands resulted in a muted ITC enthalpy profile that could not be faithfully fitted (*Figure 3—figure supplement 6*), indicating RXR ligand binding generally weakens Nur77-RXRγ LBD heterodimer interaction. Although these ITC results are qualitative, they are consistent with our NMR findings indicating that RXR ligand binding promotes dissociation of the Nur77-RXRγ LBD heterodimer to various degrees.

Finally, size exclusion chromatography (SEC) studies revealed that RXR ligands that activate Nurr1-RXRα transcription release a population of monomeric Nurr1 LBD monomer from the Nurr1-RXRα LBD heterodimer (*Yu et al., 2023*). We therefore performed SEC to assess how the RXR ligand set influences the oligomeric state of the Nur77-RXRγ LBD heterodimer (*Figure 3d*). Pharmacological RXR agonists that activate Nur77-RXRγ transcription dissociated a Nur77 LBD monomer species from the Nur77-RXRγ LBD heterodimer in the SEC profiles. Notably, the Nur77 LBD monomer species freed from the Nur77-RXRγ LBD heterodimer appears left shifted compared to the elution profile of Nur77 LBD alone. In our previous Nurr1-RXRα study, the RXR ligand-induced freed Nurr1 LBD monomer species eluted at a volume similar to the profile of Nurr1 LBD alone. Taken together, this indicates the Nur77 LBD monomer population freed by RXR ligand binding likely exchanges with the Nur77-RXRγ LBD heterodimer more rapidly than Nurr1-RXRα, which is consistent with the NMR data that shows a mixture of RXR ligand-induced exchange regimes for Nur77-RXRγ but predominantly slow exchange for Nurr1-RXRα. Interestingly, BRF110 and HX600, two compounds reported as selective agonists of

Nurr1-RXRα, also dissociated the Nur77-RXRγ LBD heterodimer and freed a population of Nur77 LBD monomer. Finally, most of the RXR antagonists did not influence the Nur77-RXRγ SEC profile, except for two compounds. UVI3003 displayed a complex SEC profile similar to what we previously observed for Nurr1-RXRα, whereas PA452, which did not dissociate Nurr1-RXRα, induced a population of freed Nur77 LBD monomer, albeit to a lesser degree than RXR agonists.

## Correlation analysis reveals the importance of a functionally diverse RXR ligand to illuminate the LBD heterodimer dissociation mechanism

Our findings indicate the RXR ligand set may affect Nur77-RXRγ transcription through classical pharmacological activation and LBD heterodimer dissociation. This prompted us to reconsider our previous observations with Nurr1-RXRα, where the lack of correlation between Nurr1-RXRα transcription and pharmacological RXRα agonism illuminated the LBD heterodimer dissociation mechanism (*Yu et al., 2023*). Close inspection of the correlation plots in our previously Nurr1-RXRα study revealed two compounds that stand out in the ligand profiling data: the Nurr1-RXRα selective agonists BRF110 and HX600. When these two Nurr1-RXRα selective agonists are excluded from the correlation analysis, a significant correlation emerges between Nurr1-RXRα transcription and features of pharmacological RXR agonism, including RXRα transcription (*Figure 4a*) and RXRα LBD coactivation interaction in the TR-FRET biochemical assay data (*Figure 4b*).

We previously used principal component analysis (PCA) to reveal in an unbiased way that Nurr1-RXRα transcriptional activation by the RXR ligand set is not correlated with pharmacological RXRα agonism (RXRα LBD TR-FRET and RXRα transcription) but instead correlated to features of LBD heterodimer dissociation (NMR and ITC data; *Yu et al., 2023*). We reperformed PCA using our previously published Nurr1-RXRα experimental data (*Figure 4c*) and found that excluding BRF110 and HX600 resulted in a correlation between Nurr1-RXRα transcriptional activation and both pharmacological RXRα agonism and LBD heterodimer dissociation (*Figure 4d*). The RXR ligand set used here does not show any selective Nur77-RXRγ transcriptional features, which would be apparent if a ligand activated Nur77-RXRγ transcription but showed no change in RXRα homodimer transcription (*Figure 2*). Consistent with the lack of selective Nurr1-RXRγ agonists in the RXR ligand set, PCA analysis revealed that Nurr1-RXRα transcriptional activation is correlated to features of pharmacological RXRα agonism and LBD heterodimer dissociation (*Figure 4e*).

## Discussion

Nur77 and its closely related NR4A subfamily member Nurr1 are constitutively active orphan nuclear receptors (*McMorrow and Murphy, 2011*; *Zhao and Bruemmer, 2010*). We recently revealed that Nurr1-RXRα activation occurs through an LBD heterodimer dissociation mechanism, which is distinct from the classical pharmacological NR activation mechanism. This led us to explore the mechanism by which RXR ligands regulate Nur77 activation through targeting its heterodimer partner, RXRγ.

Our data here, together with previous studies (*Aarnisalo et al., 2002*; *Forman et al., 1995*; *Yu et al., 2023*), show that RXRs repress transcription mediated by Nur77 and Nurr1. In our RXRγ truncation studies, the RXRγ LBD appeared important for repression of Nur77 activity, as the construct lacking the LBD (RXRγ ΔLBD) did not repress Nur77-mediated transcription. This is consistent with our published observations for Nurr1-RXRα, in which the RXRα LBD was sufficient to repress Nurr1-mediated transcription and contributed to the discovery of the RXR ligand-induced LBD heterodimer dissociation mechanism (*Yu et al., 2023*). However, unlike Nurr1-RXRα, the RXRγ LBD was not sufficient for repression of Nur77-mediated transcription. One possible explanation is suggested by the behavior of the RXRγ ΔNTD construct, which showed higher Nur77-mediated transcription than wild-type RXRγ, whereas our previous study demonstrated the RXRα ΔNTD construct showed reduced repression of Nurr1-mediated transcription (*Yu et al., 2023*). Together, these observations are consistent with a model in which the RXRγ LBD contributes to functionally relevant interaction with Nur77, while the RXRγ NTD contributes to repression. However, the present data do not definitively establish the RXRγ LBD as the only protein-protein interaction interface for Nur77, nor do they resolve the mechanism by which the RXRγ NTD influences transcription. Notably, the NTDs of RXRβ and RXRγ are capable of undergoing liquid-liquid phase separation in vitro (*Sołtys and Ożyhar, 2023*; *Sołtys and Ożyhar, 2020*), and RXRα forms biomolecular condensates in cells with another permissive

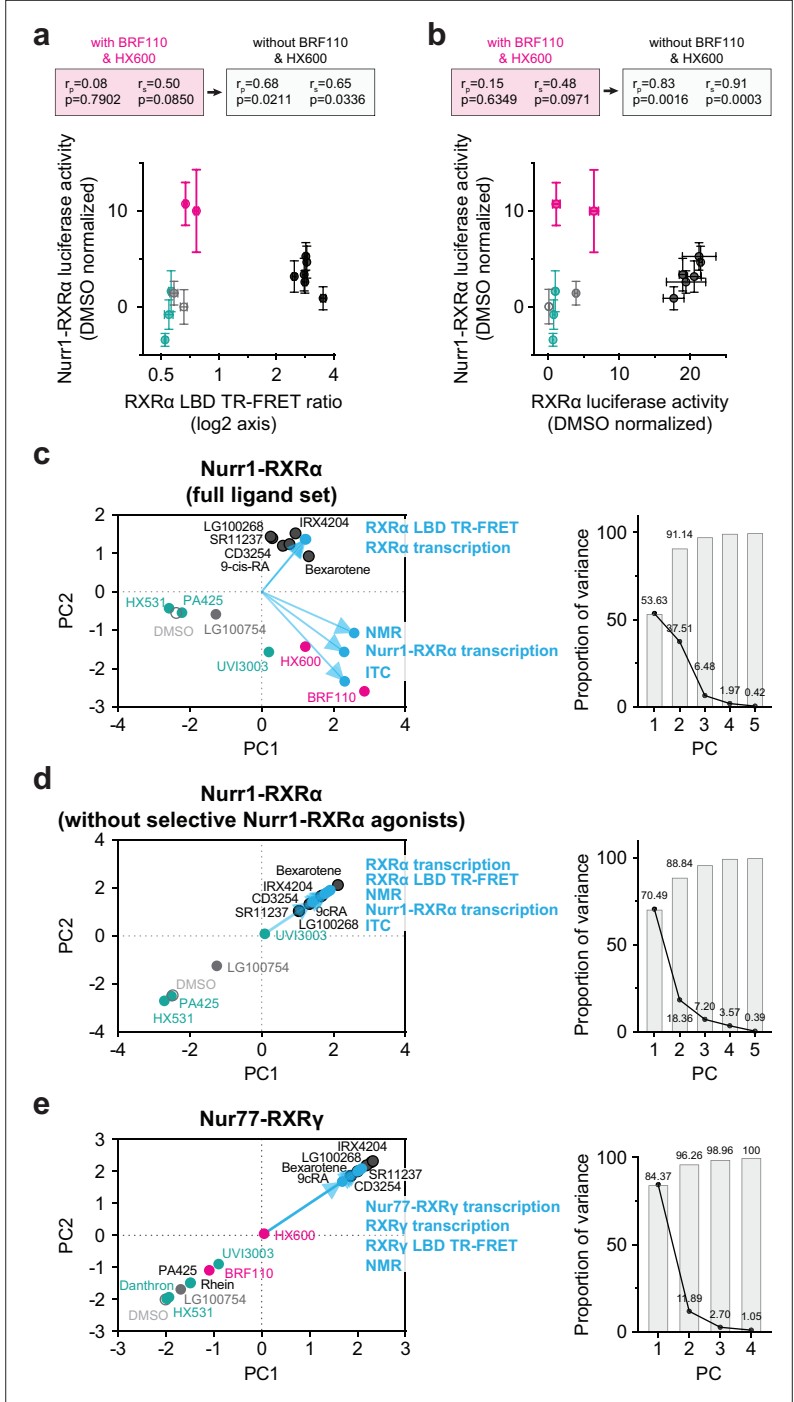

**Figure 4.** Reanalysis of published Nurr1-RXRα correlation data excluding Nurr1-RXRα selective agonists. (**a,b**) Correlation plots of our previously reported (**a**) Nurr1-RXRα cellular transcription data vs. RXRα LBD TR-FRET data and (**b**) Nurr1-RXRα cellular transcription data vs. RXRα LBD cellular transcription data. Pearson ($r_p$) and Spearman ($r_s$) correlation coefficients calculated with or without the two Nurr1-RXRα specific agonists, BRF110 and HX600 (pink data points). (**c–e**) Principal component analysis (PCA) 2D biplots (left) and proportion of variance plots (right) for our previously published Nurr1-RXRα ligand profiling data (**c**) including or (**d**) excluding the Nurr1-RXRα selective agonists BRF110 and HX600; and the (**e**) Nur77-RXRγ ligand profiling data from this study. Biplots contain the loadings (data types; blue text and blue circles) and ligand-specific PC scores of the first two PCs. Data and RXR ligand label text is colored according to RXR ligand activity as grouped in *Figure 2—figure supplement 1*.

heterodimer partner, PPARγ, to promote gene expression (*Li et al., 2022*). Thus, one possibility is that the removal of the RXRγ NTD alters condensate formation in cells or changes the physical properties of RXRγ-containing condensates, thereby influencing the repressive function observed for wild-type RXRγ on Nur77-mediated transcription. In addition, Nur77 has been observed in cellular condensates related to its functions outside of the nucleus (*Peng et al., 2021*). Future studies are needed to determine how the NTDs of these NRs influence their localization and function within biomolecular condensates and recruitment of other coregulator proteins in cells.

Our previous discovery of the RXR ligand-induced Nurr1-RXRα LBD heterodimer dissociation mechanism was possible because we were able to assemble a functionally diverse RXR ligand set that included classical pharmacological agonists and antagonists of RXR homodimers, as well as two selective Nurr1-RXRα agonists. BRF110 and HX600 compounds are unique in that they activate Nurr1-RXRα heterodimer transcription but display antagonist activity towards RXRα homodimers. Unfortunately, no selective Nur77-RXRγ agonists that activate Nur77-RXRγ transcription but antagonize RXRγ homodimer transcription have been reported. BRF110 was previously shown to activate transcription of Nurr1-RXRα heterodimer but not other NR-RXR heterodimers, including Nur77-RXRγ (*Spathis et al., 2017*), which is consistent with our data. On the other hand, data here and in our Nurr1-RXRα publication (*Yu et al., 2023*) show that HX600 functions as an agonist of Nurr1-RXRα and Nur77-RXRγ with an interesting profile for RXRα and RXRγ homodimers. HX600 partially activates RXRα and RXRγ homodimer transcription but does not enhance PGC1α coactivator peptide interaction with the RXRα LBD or RXRγ LBD in the TR-FRET assay. Although PGC1α is an important coactivator in RXR-mediated transcription (*Delerive et al., 2002*), RXRs interact with other coactivators in a ligand-dependent manner, including thyroid hormone receptor-associated proteins (TRAP) and steroid receptor coactivators (SRCs; *Johnson and O'Malley, 2012*; *Kurcinski and Kolinski, 2010*; *Malik et al., 2004*). It is possible that HX600 modulates interactions with a different set of coactivator proteins rather than PGC1α. Taken together, our findings suggest that BRF110 and HX600 may alter the conformation or interaction properties of RXRα and RXRγ LBDs in a manner that disrupts their interaction with Nurr1 and Nur77 LBDs, respectively. Additional studies into the mechanism of action of BRF110 and HX600 are warranted and may lead to a better understanding of how to elicit selective Nur77-RXRγ activation.

In summary, our published findings (*Yu et al., 2023*) and new data presented here indicate that transcriptional activation of NR4A-RXR heterodimers by RXR ligands is explained in the case of Nurr1-RXRα, and is consistent in the case of Nur77-RXRγ, with a non-classical mechanism involving LBD heterodimer dissociation.

Our findings enhance understanding of ligand-dependent NR functions, in particular regarding NR-RXR heterodimers, adding to and reaffirming the complexity and diversity of ligand-dependent NR regulatory mechanisms.

# Materials and methods

## Key resources table

| Reagent type (species) or resource | Designation | Source or reference | Identifiers | Additional information |
|---|---|---|---|---|
| Strain, strain background (*Escherichia coli*) | BL21(DE3) | Sigma-Aldrich | CMC0014 | Electrocompetent cells |
| Cell line (*Homo sapiens*) | Human embryonic kidney epithelial | ATCC | CRL-11268 | |
| Cell line (*Homo sapiens*) | SK-N-BE(2) neuroblastoma | ATCC | CRL-2271; RRID:CVCL_0528 | |
| Antibody | LanthaScreen Elite Tb-anti-His antibody | Thermo Fisher | #PV5895; RRID:AB_3720338 | |
| Chemical compound, drug | BRF110 | This study | | Synthesis procedure for BRF110 was described previously as cited in the methods. |

*Continued on next page*

*Continued*

| Reagent type (species) or resource | Designation | Source or reference | Identifiers | Additional information |
|---|---|---|---|---|
| Chemical compound, drug | HX600 | Axon Medchem | CAS 172705-89–4 | |
| Chemical compound, drug | 9-cis-Retinoic acid | Cayman Chemicals | CAS 5300-03-8 | |
| Chemical compound, drug | Bexarotene | Cayman Chemicals | CAS 153559-49–0 | |
| Chemical compound, drug | LG100268 | Cayman Chemicals | CAS 153559-76-3 | |
| Chemical compound, drug | CD3254 | Cayman Chemicals | CAS 196961-43-0 | |
| Chemical compound, drug | SR11237 | Tocris Bioscience | CAS 146670-40-8 | |
| Chemical compound, drug | UVI3003 | Cayman Chemicals | CAS 847239-17-2 | |
| Chemical compound, drug | LG100754 | Cayman Chemicals | CAS 180713-37-5 | |
| Chemical compound, drug | IRX4204 | MedChemExpress | CAS 220619-73-8 | |
| Chemical compound, drug | Rhein | Sigma-Aldrich | CAS 478–43-3 | |
| Chemical compound, drug | HX531 | Cayman Chemicals | CAS 188844-34-0 | |
| Chemical compound, drug | Danthron | Sigma-Aldrich | CAS 117-10-2 | |
| Chemical compound, drug | PA452 | Tocris Bioscience | CAS 457657-34-0 | |
| Peptide, recombinant protein | FITC-PGC1α | LifeTein | | Amino acid sequence: EAEEPSLLKKLLLAPANTQ, with a N-terminal FITC label and an amidated C-terminus. |
| Recombinant DNA reagent | Nur77-ligand binding domain (LBD) in pET45b(+) | This study | Bacteria expression plasmid | Residues: 356–598 |
| Recombinant DNA reagent | RXRγ-ligand binding domain (LBD) in pET45b(+) | This study | Bacteria expression plasmid | Residues: 233–459 |
| Recombinant DNA reagent | pET45b(+) | Novagen | 71327–3 | |
| Transfected construct (*Photinus pyralis*) | 3xNBRE-luciferase plasmid | *de Vera et al., 2016* | Sanger sequenced | |
| Transfected construct (*Photinus pyralis*) | 3xDR1-luciferase plasmid | *Hughes et al., 2014* | Mammalian expression plasmid, Sanger sequenced | This is the 3xPPRE-luciferase reporter plasmid in the referenced paper |
| Transfected construct (human) | Full-length human Nur77 in pcDNA3.1 | This study | Mammalian expression plasmid, Sanger sequenced | |

*Continued on next page*

*Continued*

| Reagent type (species) or resource | Designation | Source or reference | Identifiers | Additional information |
|---|---|---|---|---|
| Transfected construct (human) | Full-length human RXRγ in pcDNA3.1 | This study | Mammalian expression plasmid, Sanger sequenced | |
| Recombinant DNA reagent | pcDNA3.1 empty vector | Thermo Fisher Scientific | V790-20 | |
| Sequence-based reagent | RXRγ-ΔLBD-F | This paper | PCR primer ordered from Sigma | CTACCAGTGGTTAGGAAGACATG |
| Sequence-based reagent | RXRγ-ΔLBD-R | This paper | PCR primer ordered from Sigma | CATGTCTTCCTAACCACTGGTAG |
| Sequence-based reagent | ΔNTD-RXRγ | This paper | gBlock sequences ordered from IDT | gBlock sequence in *Supplementary file 1* |
| Sequence-based reagent | RXRγ-hinge-LBD | This paper | gBlock sequences ordered from IDT | gBlock sequence in *Supplementary file 1* |
| Gene (human) | Nur77 (NR4A1) | Uniprot | | Full length: residues 1–598; LBD: residues 356–598 |
| Gene (human) | RXRγ (NR2B3) | Uniprot | | Full length: residues 1–463; LBD: residues 226–462 |
| Sequence-based reagent | Restriction enzymes, ligase for cloning | NEB | | BamHI |
| Sequence-based reagent | RXRγ-ΔNTD | This paper | gBlock for Gibson assembly | CTTAAGCTTGGTACCGAGCTCGATGTGTGCTATCTG TGGAGACAGATCCTCAGGAAAGCACTACGGGGTATA CAGTTGTGAAGGCTGCAAAGGGGTTCTTCAAGAGGAC GATAAGGAAGGACCTCATCTACACGTGTCGGGATAA TAAAGACTGCCTCATTGACAAGCGTCAGCGCAACCGCT GCCAGTACTGTCGCTATCAGAAGTGCCTTGTCATGGG CATGAAGAGGGAAGCTGTGCAAGAAGAAAGACAGAG GAGCCGAGAGCGAGCTGAGAGTGAGGCAGAATGTGCT ACCAGTGGTCATGAAGACATGCCTGTGGAGAGGATTCT AGAAGCTGAACTTGCTGTTGAACCAAAGACAGAATCCT ATGGTGACATGAATATGGAGAACTCGACAAATGACCCT GTTACCAACATATGTCATGCTGCTGACAAGCAGCTTTTCA CCCTCGTTGAATGGGCCAAGCGTATTCCCCACTTCTCT GACCTCACCTTGGAGGACCAGGTCATTTTGCTTCGGGC AGGGTGGAATGAATTGCTGATTGCCTCTTTCTCCCA CCGCTCAGTTTCCGTGCAGGATGGCATCCTTCTGGC CACGGGTTTACATGTCCACCGGAGCAGTGCCCACAG TGCTGGGGTCGGCTCCATCTTTGACAGAGTCCTAAC TGAGCTGGTTTCCAAAATGAAAGACATGCAGATGGA CAAGTCGGAACTGGGATGCCTGCGAGCCATTGTACT CTTTAACCCAGATGCCAAGGGCCTGTCCAACCCCTC TGAGGTGGAGACTCTGCGAGAGAAGGTTTATGCCAC CCTTGAGGCCTACACCAACGAGAAGTATCCGGAACA GCCAGGCAGGTTTGCCAAGCTGCTGCTGCGCCTCCC AGCTCTGCGTTCCATTGGCTTGAAATGCCTGGAGCA CCTCTTCTTCTTCAAGCTCATCGGGGACACCCCCAT TGACACCTTCCTCATGGAGATGTTGGAGACCCCGCT GCAGATCACCTGAGATCCACTAGTCCAGTGTGG |

*Continued on next page*

*Continued*

| Reagent type (species) or resource | Designation | Source or reference | Identifiers | Additional information |
|---|---|---|---|---|
| Sequence-based reagent | RXRγ-hinge-LBD | This paper | gBlock for Gibson assembly | CTTAAGCTTGGTACCGAGCTCGATGAAGAGGGAAGC TGTGCAAGAAGAAAGACAGAGGAGCCGAGAGCGAGC TGAGAGTGAGGCAGAATGTGCTACCAGTGGTCATGAAG ACATGCCTGTGGAGAGGATTCTAGAAGCTGAACTTG CTGTTGAACCAAAGACAGAATCCTATGGTGACATGA ATATGGAGAACTCGACAAATGACCCTGTTACCAACATATG TCATGCTGCTGACAAGCAGCTTTTCACCCTCGTTGA ATGGGCCAAGCGTATTCCCCACTTCTCTGACCTCAC CTTGGAGGACCAGGTCATTTTGCTTCGGGCAGGGTG GAATGAATTGCTGATTGCCTCTTTCTCCCACCGCTCAGTT TCCGTGCAGGATGGCATCCTTCTGGCCACGGGTTTA CATGTCCACCGGAGCAGTGCCCACAGTGCTGGGGTC GGCTCCATCTTTGACAGAGTCCTAACTGAGCTGGTT TCCAAAATGAAAGACATGCAGATGGACAAGTCGGAA CTGGGGATGCCTGCGAGCCATTGTACTCTTTAACCCA GATGCCAAGGGCCTGTCCAACCCCTCTGAGGTGGAG ACTCTGCGAGAGAAGGTTTATGCCACCCTTGAGGCC TACACCAAGCAGAAGTATCCGGAACAGCCAGGCAGG TTTGCCAAGCTGCTGCTGCGCCTCCCAGCTCTGCGT TCCATTGGCTTGAAATGCCTGGAGCACCTCTTCTTCTTCA AGCTCATCGGGGGACACCCCCATTGACACCTTCCTCA TGGAGATGTTGGAGACCCCGCTGCAGATCACCTGAG ATCCACTAGTCCAGTGTGG |
| Commercial assay or kit | Gibson assembly | NBE | E2611L | |
| Commercial assay or kit | Britelite plus Reporter Gene Assay System | Perkin Elmer | 6066769 | |
| Software, algorithm | NITPIC software | *Keller et al., 2012* | | Baseline calculation, curve integration |
| Software, algorithm | SEDPHAT | *Brautigam et al., 2016* | | Estimation of binding affinity and thermodynamic parameter measurements |
| Software, algorithm | GUSSI | *Brautigam, 2015* | | Plot ITC figures |
| Other | NMR chemical shift assignment of Nur77 LBD | This paper | BMRB52973; https://doi.org/10.13018/BMR52973 | Published NMR peak assignment from Biological Magnetic Resonance Data Bank |
| Software, algorithm | NMRFx | *Norris et al., 2016* | | NMR data process and analysis |
| Software, algorithm | Pearsons and Spearman correlation analysis | GraphPad Prism | | Correlation analysis |
| Software, algorithm | Principal Component Analysis (PCA); | GraphPad Prism | | Correlation analysis |
| Software, algorithm | ANOVA multiple comparison test | GraphPad Prism | | Statistical testing |

## Materials and reagents

All ligands except BRF110 were obtained from commercial vendors including Cayman Chemicals, Sigma, Axon Medchem, MedChemExpress, or Tocris Bioscience: HX600 (CAS 172705-89-4), 9cRA (CAS 5300-03-8), Bexarotene (CAS 153559-49-0), LG100268 (CAS 153559-76-3), CD3254 (CAS 196961-43-0), SR11237 (CAS 146670-40-8), UVI3003 (CAS 847239-17-2), LG100754 (CAS 180713-37-5), IRX4204 (CAS 220619-73-8), Rhein (CAS 478-43-3), HX531 (CAS 188844-34-0), Danthron (CAS 117-10-2), and PA452 (CAS 457657-34-0). FITC-labeled LXXLL-containing peptide derived from human PGC-1α (137-155; EAEEPSLLKKLLLAPANTQ) was synthesized by LifeTein with an N-terminal FITC label and an amidated C-terminus for stability. Bacterial expression plasmids included human Nur77 (NR4A1) LBD (residues 356–598) inserted into a pET45b(+) plasmid (Novagen) as a 3C-cleavable N-terminal hexahistidine (6xHis)-tag fusion protein; and human RXRγ (NR2B3) LBD (residues 226–463) inserted into a pET45b(+) plasmid (Novagen) as a TEV-cleavable N-terminal hexahistidine (6xHis)-tag fusion protein. Luciferase reporter plasmids included a 3xNBRE-luciferase plasmid

containing three copies of the NGFI-B response element corresponding to the monomeric binding site for Nur77 (*de Vera et al., 2016*; *Murphy et al., 1996*); and a 3xDR1-luciferase containing three copies of the optimal direct repeat 1 (DR1) binding site for RXRγ homodimers (*Hughes et al., 2014*; *Osz et al., 2015*; *Subauste et al., 1994*). Mammalian expression plasmids included full-length human Nur77 (residues 1–598) and full-length human RXRγ (residues 1–463) subcloned into a pcDNA3.1 vector. To clone the RXRγ ΔLBD construct (residues 230–463), site-directed mutagenesis and PCR were used to insert a stop codon before the start of the LBD using the full-length RXRα expression plasmid the following primers: forward primer, CTACCAGTGGTTAGGAAGACATG; reverse primer, CATGTCTTCCTAACCACTGGTAG. To clone the ΔNTD RXRγ construct (residues 139–463) and RXRγ-hinge-LBD construct (205-463), BamHI was used to cut pcDNA3.1 and subsequently Gibson assembly was used to clone the LBD into the linearized vector using the gBlock sequences in *Supplementary file 1*.

## Compound synthesis

BRF110—which was commercially available at the time of our Nurr1-RXRα study (*Yu et al., 2023*) but not commercially available for this current study—was synthesized using a method similar to one that previously described the original synthesis and characterization of BRF110 (*Asvos et al., 2025*; *Spathis et al., 2017*), which is briefly described below.

Ethyl 2-(2,2,2-trifluoroacetyl)pent-4-enoate (**2**). Ethyl 4,4,4-trifluoroacetoacetate (**1**, 28.26 g, 153.5 mmol) was added dropwise to a suspension of NaH (6.14 g, 153.5 mmol) in THF (150 mL) at 0 °C. It was stirred for 1 h at 0 °C. The solvent was removed, and anhydrous acetone (70 mL) was added. Subsequently, potassium iodide (2.55 g, 15.36 mmol) was added followed by the dropwise addition of allyl bromide (13 ml, 150 mmol) in anhydrous acetone (40 mL). It was stirred at 60 °C for 48 hr. The solvent was removed, 1 N HCl was added, and it was extracted with DCM (2 X). The combined organic layers were dried over $Na_2SO_4$, filtered, and concentrated under reduced pressure to give the crude product without further purification.

**Chemical structure 1.** Synthesis of 5-allyl-2-phenyl-6-(trifluoromethyl)pyrimidin-4-ol (BRF110).

5-allyl-2-phenyl-6-(trifluoromethyl)pyrimidin-4-ol (**3**). Ethyl 2-(2,2,2-trifluoroacetyl)pent-4-enoate (**2**, 33.69 g, 150.28 mmol) was added to a solution of NaOEt (21% w/w in EtOH, 62 ml, 166.07 mmol) and benzamidine hydrochloride (23.54 g, 150.31 mmol). It was stirred at 100 °C for 16 hr. The solvent was removed under reduced pressure. It was acidified with 1 N HCl and extracted with DCM (2 X). It was precipitated from ethanol to yield the title compound.

5-Allyl-4-chloro-2-phenyl-6-(trifluoromethyl)pyrimidine (**4**). 5-allyl-2-phenyl-6-(trifluoromethyl)pyrimidin-4-ol (**3**, 5.4 g, 19.27 mmol) was dissolved in $POCl_3$ (20 mL). It was stirred at 125 °C for 2 hr. The solvent was removed under reduced pressure, the residue was dissolved in EtOAc, washed with saturated $NaHCO_3$ and brine. The combined organic layers were dried over $Na_2SO_4$, filtered, and concentrated under reduced pressure to give the crude product, which was purified by flash chromatography on silica gel with EtOAc/Hex (1:10) to obtain the title compound.

Methyl 4-((5-allyl-2-phenyl-6-(trifluoromethyl)pyrimidin-4-yl)amino)benzoate (**5**). 5-Allyl-4-chloro-2-phenyl-6-(trifluoromethyl)pyrimidine (**4**, 259 mg, 0.87 mmol) was dissolved in isopropanol (3 mL). A few drops of concentrated aqueous HCl were added. It was stirred at 140 °C for 48 hr. The solvent was removed under reduced pressure; the residue was dissolved in EtOAc, washed with saturated NaHCO$_3$ and brine. The combined organic layers were dried over Na$_2$SO$_4$, filtered, and concentrated under reduced pressure to give the crude product, which was purified by flash chromatography on silica gel with EtOAc/Hex to obtain the title compound.

Methyl 4-((5-allyl-2-phenyl-6-(trifluoromethyl)pyrimidin-4-yl)(methyl)amino)benzoate (**6**). Cesium carbonate (86 mg, 0.26 mmol) was added to a solution of methyl 4-((5-allyl-2-phenyl-6-(trifluoromethyl)pyrimidin-4-yl)amino)benzoate (**5**, 81 mg, 0.22 mmol) in DMF (1 mL), followed by the addition of iodomethane (47 mg, 0.33 mmol). It was stirred for 16 hr at room temperature. It was diluted with EtOAc, washed with saturated NaHCO$_3$ and brine. The combined organic layers were dried over Na$_2$SO$_4$, filtered, and concentrated under reduced pressure. The residue was purified by silica gel chromatography to yield the title compound.

4-((5-allyl-2-phenyl-6-(trifluoromethyl)pyrimidin-4-yl)(methyl)amino)benzoic acid (**BRF110**). Methyl 4-((5-allyl-2-phenyl-6-(trifluoromethyl)pyrimidin-4-yl)(methyl)amino)benzoate (**6**, 111 mg, 0.26 mmol) was dissolved in methanol (2 mL); 1 N LiOH (0.57 mL) was added. It was stirred at room temperature overnight. It was diluted with EtOAc, washed with 1 N HCl and brine. The combined organic layers were dried over Na$_2$SO$_4$, filtered, and concentrated under reduced pressure. The residue was purified by silica gel chromatography to yield the title compound. $^1$H NMR (400 MHz, d6-DMSO) δ 12.87 (s, 1 H), 8.44–8.38 (m, 2 H), 7.94–7.90 (m, 2 H), 7.60–7.54 (m, 3 H), 7.25–7.21 (m, 2 H), 5.65–5.55 (m, 1 H), 4.93 (dd, $J$=1.4, 10.2 Hz, 1 H), 4.72 (dd, $J$=1.6, 17.2 Hz, 1 H), 3.59 (s, 3 H), 3.00 (d, $J$=5.6 Hz); $^{13}$C NMR (100 MHz, d6-DMSO) δ 166.70, 165.18, 161.16, 153.31 (q, $J$=32.0), 149.91, 135.92, 133.70, 131.39, 130.96, 128.79, 127.78, 126.14, 121.34, 119.85, 116.44, 40.75, 29.99; $^{19}$F NMR (376.50 MHz, d6-DMSO) δ –62.89; HRMS (ESI) calculated for C$_{22}$H$_{19}$F$_3$N$_3$O$_2$ [M + H$^+$] 414.1429, found 414.1439. LCMS confirmed the purity was >95%.

## Cell lines for mammalian cell culture

SK-N-BE(2)-C (#CRL-2268) cells were obtained from and authenticated by ATCC, determined to be mycoplasma free, and cultured according to ATCC guidelines at low passage number (less than 10 passages; typically passages 2–4). Briefly, SK-N-BE(2)-C were grown at 37 °C and 5% CO$_2$ in a media containing a 1:1 mixture of EMEM (ATCC) and F12 medium (Gibco) supplemented with 10% fetal bovine serum (Gibco) until 90–95% confluence in T-75 flasks prior to subculture or use.

## Protein expression and purification

Proteins were expressed in *Escherichia coli* BL21(DE3) cells using terrific broth (TB) media, autoinduction ZY media, or M9 minimal media (using $^{15}$NH$_4$Cl and H$_2$O-based media for $^{15}$N-labeled protein; or $^{15}$NH$_4$Cl, $^{13}$C-glucose, and D$_2$O-based media for $^{2}$H,$^{13}$C,$^{15}$N-labeled protein). For TB expression of Nur77 LBD, cells were grown at 37 °C and induced with 1.0 mM isopropyl β-D-thiogalactoside at OD (600 nm) of 1.0, grown for an additional 16 hr at 18 °C, and then centrifuged for harvesting. For autoinduction expression of RXRγ LBD, cells were grown for 5 hr at 37 °C, then 16 hr at 22 °C for RXRγ LBD, then centrifuged for harvesting. For M9 expression of $^{15}$N-labeled Nur77 LBD and $^{2}$H,$^{13}$C,$^{15}$N-labeled Nur77 LBD, cells were grown at 37 °C and induced with 1.0 mM isopropyl β-D-thiogalactoside at OD (600 nm) of 0.6, grown for an additional 16 hr at 18 °C, and then centrifuged for harvesting. Cell pellets were lysed using sonication, and proteins were purified using Ni-NTA affinity chromatography and gel filtration/SEC. The purified proteins were verified by SDS-PAGE, then stored in a buffer consisting of 20 mM potassium phosphate (pH 7.4), 50 mM potassium chloride, and 0.5 mM EDTA. All studies that used RXRγ LBD protein used pooled SEC fractions for the apo-homodimeric form.

## Transcriptional reporter assays

SK-N-BE(2)-C cells were seeded in 9.5 cm$^2$ cell culture well (Corning) at 0.5 million for transfection using Lipofectamine 2000 (Thermo Fisher Scientific) and Opti-MEM with an empty vector control or full-length Nur77 expression plasmid (2 µg), with or without full-length or different RXRγ truncation constructs (4 µg), and 3xNBRE-luc (6 µg). After incubation for 16–20 hr, cells were transferred to white 384-well cell culture plates (Thermo Fisher Scientific) at 0.5 million cells/mL in 20 µL total volume/well.

After a 4 hr incubation, cells were treated with 20 µL of vehicle control (DMSO) or 1 µM ligand. After a final 16–20 hr incubation, cells were harvested with 20 µL Britelite Plus (PerkinElmer), and luminescence was measured on a BioTek Synergy Neo multimode plate reader. The luminescence readouts were normalized to cells transfected with the empty vector (truncation construct assay) or cells treated with DMSO (assays with or without ligand treatment). Data were plotted using GraphPad Prism. In *Figure 1*, statistical testing was performed, and p-values were calculated in GraphPad Prism using the Brown-Forsythe and Welch multiple comparisons test (does not assume equal s.d. values among compared conditions) relative to full-length Nur77 condition. Data are representative of two or more independent experiments (n=6 or 9 biological replicates, as indicated in figure legends).

## Time-resolved fluorescence resonance energy transfer (TR-FRET) assay

Assays were performed in 384-well black plates (Greiner) using 22.5 µL final well volume. Each well contained 4 nM 6xHis-RXRγ LBD, 1 nM LanthaScreen Elite Tb-anti-His antibody (Thermo Fisher #PV5895), and 400 nM FITC-labeled PGC1α peptide in a buffer containing 20 mM potassium phosphate (pH 7.4), 50 mM KCl, 5 mM TCEP, and 0.005% Tween 20. Ligand stocks were prepared via serial dilution in DMSO and added to wells (10 µM final concentration) in triplicate. The plates were read using BioTek Synergy Neo multimode plate reader after incubation at 4 °C for at least 2 hr. The Tb donor was excited at 340 nm, and its emission was measured at 495 nm; the emission of the acceptor FITC was measured at 520 nm. Data were plotted using GraphPad Prism as TR-FRET ratio (measurement at 520 nm/measurement at 495 nm) at a fixed ligand concentration (2–4 µM depending on the starting compound stock concentration). Data are representative of two or more independent experiments (n=3 biological replicates).

## Size exclusion chromatography with multi-angle light scattering (SEC-MALS)

Gel filtration purified Nurr1 LBD, Nur77 LBD, RXRα LBD, and RXRγ LBD were analyzed by SEC-MALS using an HPLC system (Agilent Technologies 1260 Infinity) connected to a MALS system (Wyatt DAWN HELEOS II Ambient with Optilab TrEX HC differential refractive index detector). The SEC-MALS system was calibrated with bovine serum albumin (BSA) before the measurement. Protein was loaded onto a pre-equilibrated SEC analytical column (WTC-050S5, Wyatt) with buffer containing 40 mM potassium phosphate (pH 7.4), 200 mM KCl. An aliquot of 100 µL of each protein at a concentration of 5 mg/mL was injected with a flow rate of 0.5 ml/min. All experiments were performed at room temperature (25 °C). Data collection and SEC-MALS analysis were performed with ASTRA 8.0.0.19 (64-bit, Wyatt Technology).

## Isothermal titration calorimetry

Experiments were performed using a TA instrument affinity ITC. All experiments were solvent-matched and contained 0.25% DMSO (ligand vehicle) final concentration. RXRγ LBD was preincubated with two equivalents of vehicle (DMSO) or ligand at 5 °C. ITC experiments were performed in duplicate by titrating Nur77 LBD to RXRγ LBD in a 10:1 ratio (500 µM Nur77 LBD to 50 µM RXRγ LBD). NITPIC software (*Keller et al., 2012*) was used to calculate ITC data baselines, integrate curves, prepare experimental data for fitting in SEDPHAT (*Brautigam et al., 2016*) to obtain binding affinities and thermodynamic parameter measurements, in the case of studies performed without RXR ligand, using a homodimer competition model A+B + C <->AB + C <->AC + B; competing B and C for A, where A and C are RXRγ LBD monomer and B is Nur77 LBD monomer. Addition of RXR ligands resulted in an apparent weakening of the Nur77 LBD interaction with RXRγ LBD, which was apparent in the significantly changed and muted enthalpic response that could not be faithfully fit to obtain binding affinities. Final figures were exported using GUSSI (*Brautigam, 2015*).

## NMR spectroscopy

For NMR chemical shift assignment, $^2$H,$^{13}$C,$^{15}$N-labeled Nur77 LBD was used to collect two-dimensional (2D) and three-dimensional (3D) backbone assignment data using TROSY-based HSQC, HNCO, HN(CA)CO, HNCA, HN(CO)CA, HN(CA)CB, HN(COCA)CB, and CC(CO)NH-TOCSY experiments performed at 298 K on a Bruker 700 MHz NMR instrument equipped with a QCI cryoprobe using Bruker Topspin (version 2). NMR assignments were performed using RunAbout in NMRViewJ

(*Johnson, 2018*; *Johnson, 2004*). For NMR structural footprinting studies, two-dimensional (2D) [$^1$H,$^{15}$N]-TROSY-HSQC NMR experiments were performed at 298 K on a Bruker 900 MHz NMR instrument equipped with a TCI cryoprobe. Samples were prepared in a buffer containing 20 mM potassium phosphate (pH 7.4), 50 mM potassium chloride, 0.5 mM EDTA, and 5% dimethyl sulfoxide-d$_6$ (DMSO-d$_6$). Data were collected using Bruker Topspin (version 2) using 200 µM $^{15}$N-labeled Nur77 LBD with or without two molar equivalents of unlabeled RXRγ LBD in the absence or presence of RXR ligands. Data were processed and analyzed using NMRFx (version 11.4 .x; *Norris et al., 2016*) using unpublished NMR chemical shift assignments for the Nur77 LBD obtained from our lab. Relative Nur77 LBD monomer populations were estimated by the relative peak intensities of the monomeric ($I_{monomer}$) and heterodimer ($I_{heterodimer}$) species using the following equation: $I_{monomer\_population} = I_{monomer}/(I_{monomer} + I_{heterodimer})$.

## Analytical SEC

To prepare Nur77-RXRγ LBD heterodimer for analytical SEC analysis, purified Nur77 LBD and RXRγ LBD (each 700 µM in 2.5 mL) were incubated together at 4 °C in their storage buffer containing 20 mM potassium phosphate (pH 7.4), 50 mM potassium chloride, and 0.5 mM EDTA for 16 hr, injected (5 mL total) into HiLoad 16/600 Superdex 75 pg (Cytiva) connected to an AKTA FPLC system (GE Healthcare Life Science). Gel filtration was performed using the same buffer to purify Nur77-RXRγ LBD heterodimer for analysis, collecting 2 mL fractions to isolate heterodimer population into 15 samples of 0.5 mL (at a protein concentration of 300 µM) for analytical gel filtration. Additionally, purified Nur77 LBD and RXRγ LBD pooled gel filtration fractions consisting of the homodimeric species were also used. Samples containing RXRγ LBD (homodimer or heterodimer with Nur77 LBD) were incubated with two molar equivalents of each ligand for 16 hr at 4 °C before injection onto Superdex 75 Increase 10/300 GL (Cytiva) connected to the same AKTA FPLC system. The UV chromatograms were exported in CSV format and plotted.

## AlphaFold3 model of Nur77-RXRγ

A structural model of the human Nur77-RXRγ LBD heterodimer was generated using AlphaFold3 webserver (https://alphafoldserver.com) (*Abramson et al., 2024*) using the same sequences from our bacterial protein expression constructs. PyMOL (version 3) was used for structural visualization and plotting.

## Correlation and other statistical analyses

Correlation plots were performed using GraphPad Prism to calculate Pearson ($r_p$) and Spearman ($r_s$) correlation coefficients and two-tailed p-values between two experimental measurements per plot. Statistical testing of control to variable conditions was performed using one-way ANOVA testing as detailed in the figure legends. PCA was performed in GraphPad Prism; all experimentally determined data were used as variables (method = standardize, PCs selected based on parallel analysis at the 95% percentile level with 1000 simulations). p-value statistical significance shorthand, where present, conforms to GraphPad Prism standards: n.s., p>0.05; *, p≤0.05; **, p≤0.01; ***, p≤0.001; and ****, p≤0.0001.

## Acknowledgements

This work was supported in part by the National Institutes of Health (NIH) grant R01AG070719 from the National Institute of Aging (NIA) and grants for NMR instrumentation from the NSF-MRI (0922862) resulting in the acquisition of a 900 MHz Ultra-High Field NMR spectrometer in 2009; NIH S10RR025677 for console upgrades on all biomolecular NMR spectrometers in 2009; NIH R35GM118089-04S1 supplement for a helium liquefier in 2019; NIH S10OD034276 to replace the 800 MHz spectrometer in 2024, accompanied by Vanderbilt University matching funds. The contents of this publication are solely the responsibility of the authors and do not necessarily represent the official views of NIH or NSF.

## Additional information

### Funding

| Funder | Grant reference number | Author |
|---|---|---|
| National Institute on Aging | R01AG070719 | Douglas J Kojetin |

The funders had no role in study design, data collection and interpretation, or the decision to submit the work for publication.

### Author contributions

Xiaoyu Yu, Conceptualization, Formal analysis, Validation, Investigation, Visualization, Methodology, Writing – original draft, Writing – review and editing; Yuanjun He, Investigation, Writing – review and editing; Thedore M Kamenecka, Supervision, Writing – review and editing; Douglas J Kojetin, Conceptualization, Formal analysis, Supervision, Funding acquisition, Validation, Visualization, Writing – original draft, Project administration, Writing – review and editing

### Author ORCIDs

Xiaoyu Yu (ID) https://orcid.org/0000-0003-0549-9560
Douglas J Kojetin (ID) https://orcid.org/0000-0001-8058-6168

Reviewer #1 (Public review): https://doi.org/10.7554/eLife.106861.3.sa1
Reviewer #2 (Public review): https://doi.org/10.7554/eLife.106861.3.sa2
Author response https://doi.org/10.7554/eLife.106861.3.sa3

## Additional files

### Supplementary files

Supplementary file 1. gBlock sequences used to generate RXRγ truncation constructs.

Supplementary file 2. AlphaFold3 model of the Nur77-RXRγ LBD.

MDAR checklist

### Data availability

NMR chemical shift assignments have been deposited in the Biological Magnetic Resonance Bank (BMRB) under accession code 52973. AlphaFold3 model of the Nur77-RXRγ LBD is available as *Supplementary file 2*. All other data generated or analyzed during this study are included in the manuscript as source data files or, in the case of Figure 4a-d was previously published (*Yu et al., 2023*).

The following dataset was generated:

| Author(s) | Year | Dataset title | Dataset URL | Database and Identifier |
|---|---|---|---|---|
| Yu X, Kojetin D | 2025 | 1H,15N.13C NMR backbone assignments for human Nur77 ligand-binding domain (LBD) | https://bmrb.io/data_library/summary/index.php?bmrbId=52973 | Biological Magnetic Resonance Bank, 52973 |

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
