## [Editor Report · eLife Assessment]

This **important** study investigated whether the nuclear receptor Nur77 is regulated by a non-canonical mechanism of ligand-induced disruption of its interaction with RXRg, similar to the family member Nurr1. The overall evidence is **compelling**. This manuscript will be of interest to scientists focusing on mechanisms of transcriptional regulation.

---

## [Referee Report · Reviewer #1 (Public review)]

Summary:

This foundational study builds on prior work from this group to reveal the complexities underlying ligand-dependent RXRγ-Nur77 heterodimer formation, offering a compelling re-evaluation of their earlier conclusions. The Authors examine how a library of RXR ligands influences the biophysical, structural, and functional properties of Nur77. They find that although the Nur77-RXRγ heterodimer shares notable functional similarities with the Nurr1-RXRα complex, it also exhibits unique features - notably, both dimer dissociation and classical agonist-driven activities. This work advances our understanding of the nuanced behaviors of nuclear receptor heterodimers, which have important implications for health and disease.

Strengths:

(1) Builds on previous work by providing a comprehensive analysis that examines whether Nur77-RXRγ heterodimer formation parallels that of the Nurr1-RXRα complex.

(2) Systematic evaluation of a library of RXR ligands provides a broad survey of functional outputs.

(3) Careful reanalysis of previous work sheds new light on how NR4A heterodimers function.

Weaknesses:

(1) Some conclusions appear overstated or are not well substantiated by the work presented. It's unclear how the data support a non-classical mode of agonism, for example, based on the data shown.

(2) Some assays have relatively few replicates, with only two in some cases.

Comments on revisions:

I'm satisfied with the revised version.

---

## [Referee Report · Reviewer #2 (Public review)]

Summary:

This study explores the mechanisms by which binding of the nuclear receptor RXRg regulates its heterodimeric partner Nur77. Previously, this group made the interesting discovery that ligand-dependent activation of RXRg bound to a related partner, Nurr1, does not occur through a classical pharmacological mechanism but through agonist-dependent dissociation of the complex through disruption of their ligand binding domain (LBD) interactions. Here, they revisit this paradigm with Nur77. In contrast to Nurr1, the authors do not have the reagents to clearly support a role for LBD dissociation. Following from the model of partial ligand-dependent dissociation of the LBD heterodimer, the experimental data (NMR, ITC, SEC) are interesting and quite complex.

Strengths:

The authors do a rigorous job of describing the data and providing possible interpretations and caveats. Revisiting the analysis of Nurr1, they identify the crucial role that selective Nurr1-RXRg agonists played in supporting the LBD dissociation model; without analogous compounds for the Nur77-RXRg complex, it is difficult to invoke this mechanism. Interestingly, treatment with the Nurr1-RXRg selective agonist HX600 suggests it can induce some LBD dissociation. Therefore, there may be some similarities between regulation of Nurr1 and Nur77 by RXRg.

Weaknesses:

Despite evidence supporting a partial role for RXRg LBD dissociation as a mechanism to activate Nur77, other data demonstrate that a fundamentally different regulatory mechanism likely exists in the Nur77-RXRg complex that involves the RXRg disordered NTD. The decision to describe further study of this as outside the scope of this work is unfortunate, as it closed off an avenue that could have provided fruitful data informing the apparently distinct regulatory mechanisms of the Nur77-RXRg complex. Given the uncertainty in the importance of the partial roles of the pharmacological mechanism, LBD dissociation, and the RXRg NTD, this study may have limited impact on the field.

Comments on revisions:

I'm satisfied with the revision.

---

## [Author Response]

The following is the authors’ response to the original reviews

**Public Reviews:**

**Reviewer #1 (Public review):**
Summary:This foundational study builds on prior work from this group to reveal the complexities underlying ligand-dependent RXRγ-Nur77 heterodimer formation, offering a compelling re-evaluation of their earlier conclusions. The authors examine how a library of RXR ligands influences the biophysical, structural, and functional properties of Nur77. They find that although the Nur77-RXRγ heterodimer shares notable functional similarities with the Nurr1-RXRα complex, it also exhibits unique features, notably, both dimer dissociation and classical agonist-driven activities. This work advances our understanding of the nuanced behaviors of nuclear receptor heterodimers, which have important implications for health and disease.Strengths:(1) Builds on previous work by providing a comprehensive analysis that examines whether Nur77-RXRγ heterodimer formation parallels that of the Nurr1-RXRα complex.(2) Systematic evaluation of a library of RXR ligands provides a broad survey of functional outputs.(3) Careful reanalysis of previous work sheds new light on how NR4A heterodimers function.

We thank the reviewer for recognizing our work as foundational. In the nuclear receptor field, current understanding of ligand-regulated nuclear receptor activity is based largely on ligand-dependent coregulator recruitment preferences; for example, agonists enhance coactivator recruitment to activate transcription. Building on our recent study of Nurr1-RXRα, the present work suggests that activation of the evolutionarily related NR4A-RXR heterodimer Nur77-RXRγ by RXR ligands is also consistent with a non-classical activation mechanism involving heterodimer dissociation.

Weaknesses:(1) Some conclusions appear overstated or are not well substantiated by the work presented. It's unclear how the data support a non-classical mode of agonism, for example, based on the data shown.

We thank the reviewer for this important point. We did not intend to claim that Nur77-RXRγ activation is explained exclusively by a non-classical mode of agonism. Rather, our interpretation was that the data are consistent with two possible, non-mutually exclusive mechanisms: (1) a classical pharmacological mechanism involving ligand-dependent coregulator recruitment; and (2) a non-classical mechanism involving ligand-binding domain (LBD) heterodimer dissociation, as we previously described for Nurr1-RXRα. This differs from our prior eLife study of Nurr1-RXRα, in which the data supported the LBD heterodimer dissociation model but not the classical pharmacological model.

In our revised manuscript, we clarify two points that are important for interpreting the Nur77-RXRγ data. First, several experimental limitations of the Nur77-RXRγ studies reduced the extent to which the mechanism could be resolved as rigorously as in our earlier Nurr1-RXRα study. Second, and more importantly, the currently available ligand set lacks Nur77-RXRγ-selective agonists. This limits our ability to determine whether LBD heterodimer dissociation is the sole or principal mechanism of activation, or instead one of several contributing mechanisms.

Taken together, these results support LBD heterodimer dissociation as a plausible and experimentally observable component of Nur77-RXRγ activation and, therefore, as a candidate shared activation mechanism for NR4A-RXR heterodimers. At the same time, because the quantitative evidence is less definitive than in the Nurr1-RXRα system, we agree that conclusions regarding Nur77-RXRγ should be stated more cautiously. This caution is reflected in both the title of our manuscript (“Towards a unified mechanism…”) and the language used throughout the text.

(2) Some assays have relatively few replicates, with only two in some cases.

We thank the reviewer for their attention to experimental rigor. For some assays, the findings were reproduced in two independent experiments, which we considered sufficient to confirm the presence and reproducibility of the effects observed in those particular assay formats. In the original manuscript, we used a general statement in the figure legends (“representative of two or more independent experiments”) across all assay data. In the revised manuscript, we now specify the number of independent experimental replicates for each assay in the corresponding figure legends to improve transparency.

**Reviewer #2 (Public review):**
Summary:This study explores the mechanisms by which binding of the nuclear receptor RXRg regulates its heterodimeric partner Nur77. Previously, this group made the interesting discovery that ligand-dependent activation of RXRg bound to a related partner, Nurr1, does not occur through a classical pharmacological mechanism but through agonist-dependent dissociation of the complex through disruption of their ligand binding domain (LBD) interactions. Here, they revisit this paradigm with Nur77. In contrast to Nurr1, the authors do not have the reagents to clearly support a role for LBD dissociation. Following the model of partial ligand-dependent dissociation of the LBD heterodimer, the experimental data (NMR, ITC, SEC) are interesting and quite complex.Strengths:The authors do a rigorous job of describing the data and providing possible interpretations and caveats. Revisiting the analysis of Nurr1, they identify the crucial role that selective Nurr1-RXRg agonists played in supporting the LBD dissociation model; without analogous compounds for the Nur77-RXRg complex, it is difficult to invoke this mechanism. Interestingly, treatment with the Nurr1-RXRg selective agonist HX600 suggests it can induce some LBD dissociation. Therefore, there may be some similarities between the regulation of Nurr1 and Nur77 by RXRg.

We thank the reviewer for this thoughtful and balanced summary of our work. We appreciate the reviewer’s recognition of both our prior findings in the Nurr1-RXRα system and the interesting, but more complex, experimental behavior observed here for Nur77-RXRγ. We agree that the absence of Nur77-RXRγ-selective agonists currently limits how definitively the contribution of LBD dissociation can be resolved, and we have revised the manuscript to make this point more explicit and to further temper our conclusions accordingly.

Weaknesses:Despite evidence supporting a partial role for RXRg LBD dissociation as a mechanism to activate Nur77, other data demonstrate that a fundamentally different regulatory mechanism likely exists in the Nur77-RXRg complex that involves the RXRg disordered NTD. The decision to describe further study of this as outside the scope of this work is unfortunate, as it closed off an avenue that could have provided fruitful data informing the apparently distinct regulatory mechanisms of the Nur77-RXRg complex. Given the uncertainty in the importance of the partial roles of the pharmacological mechanism, LBD dissociation, and the RXRg NTD, this study may have limited impact on the field.

We thank the reviewer for this thoughtful point. We agree that the RXRγ NTD likely contributes to regulation of Nur77-RXRγ transcription, and that our truncation data suggest that regions outside the LBD can influence transcriptional output. At present, however, the effect of RXRγ NTD truncation is not sufficiently mechanistically resolved to distinguish among several plausible explanations.

For example, the RXRγ NTD has been implicated in phase separation and biomolecular condensate formation in cells (PubMed ID 40392852, 40420113, 33971237, 31881311), and perturbing these properties (via RXRγ NTD truncation) could indirectly affect Nur77-RXRγ transcriptional activity. In addition, NTDs of nuclear receptors can participate in coactivator or corepressor interactions (PubMed ID 24284822), raising the possibility that removal of the RXRγ NTD alters transcription by changing recruitment of regulatory factors rather than by directly informing the LBD-centered mechanism examined here. We will clarify in the revised manuscript that these possibilities remain unresolved and represent important directions for future study.

We also agree that defining how multiple RXRγ domains contribute to Nur77-RXRγ regulation would be valuable for the field. However, the focus of the present study is narrower: to test whether, as in our previous eLife study of Nurr1-RXRα, RXR ligands can influence heterodimer function through effects on LBD-LBD interactions. Because the available data do not yet allow a mechanistic dissection of the RXRγ NTD contribution, we believe that a definitive analysis of this question would require a separate set of experiments beyond the scope of the present work. We have revised the manuscript to better acknowledge this limitation and to frame the conclusions accordingly.

**Recommendations for the authors:**

**Reviewer #1 (Recommendations for the authors):**
Overall, this is a compelling body of work. Additional summary statements and clearer transitions would be helpful throughout.Here are some points that should be addressed or at least discussed by the authors:(1) It is unclear in the luciferase assays whether the truncated proteins are functional or not. Were there Western blots or other assays run to confirm protein concentrations?

We thank the reviewer for this point. We did not perform Western blotting or other assays to confirm equivalent expression levels of the truncated RXRγ constructs, and we agree that this is a limitation of the luciferase assay data. As a result, the transcriptional effects observed with the truncation constructs should be interpreted cautiously.

With that said, the increased transcriptional activity observed upon deletion of the RXRγ NTD/AF-1 region suggests that this region may exert a repressive effect on Nur77-RXRγ transcription. This effect could reflect multiple, non-mutually exclusive mechanisms, including altered phase separation or condensate-related properties of RXRγ, or altered recruitment of transcriptional coregulators through the NTD. Because our truncation strategy does not distinguish among these possibilities, we do not believe these data allow a definitive mechanistic interpretation of the NTD contribution.

We have revised the manuscript to clarify this limitation. We also note that the primary focus of the present study is the role of ligands in modulating Nur77-RXRγ function through LBD-mediated interactions, in direct comparison with our previous Nurr1-RXRα study. A more complete mechanistic dissection of how RXRγ domain architecture influences Nur77-RXRγ transcription will require future work.

(2) Why does the Nur77 construct lacking the NTD show increased luciferase activity?

Please see our response above to Reviewer 2’s Public Review, which also addresses this point.

(3) A case is made for the Nur77 LBD driving the activity, but it also could be inferred that the DBD is driving based on the data shown in Figure 1.

We thank the reviewer for this point. We agree that the Nur77 DBD is required for binding to NBRE response elements, and we did not intend to suggest otherwise. The experimental approach in Figure 1 was not designed to dissect the relative contributions of Nur77 domains, since Nur77 was tested only in its full-length form. Instead, the purpose of this experiment was to examine how truncation of RXRγ domains affects Nur77-RXRγ transcriptional activity, in direct comparison with our prior eLife study of Nurr1-RXRα, where RXRα domain truncations helped define the importance of RXR-LBD-mediated regulation. We will revise the text to clarify that Figure 1 does not distinguish whether Nur77 DBD-dependent DNA binding is necessary, but instead addresses whether the pattern of RXRγ domain dependence is consistent with an LBD-centered mechanism of ligand-regulated heterodimer function.

(4) It is stated that the HX600 coactivator recruitment requires further study. Why wasn't it studied here?

We thank the reviewer for this point. The primary focus of this study was to determine how RXR ligands influence Nur77-RXRγ heterodimer activity, particularly in relation to ligand-dependent effects on heterodimer function. A more detailed analysis of HX600-dependent coactivator recruitment would require a broader mechanistic investigation of RXRα and RXRγ homodimer pharmacology and RXR-specific coregulator interactions, which extends beyond the central scope of the present manuscript. We agree that this is an important question and view it as a valuable direction for future work.

(5) Figure 3B, the shifts in monomer populations, error bars aren't shown, the biggest shift is from 0.2 to 0.6, is that statistically meaningful?

We thank the reviewer for this point. The reviewer is correct that error bars were not shown for Figure 3B. These NMR measurements were performed once (n=1), and therefore the shifts in monomer populations shown in Figure 3B cannot be assessed statistically. Because these studies required substantial NMR instrument time and isotopically labeled protein at high concentration, we were not able to perform experimental replicates for this dataset. We have revised the figure legend to explicitly state that these data were collected from a single experiment and have tempered the corresponding language in the manuscript accordingly.

(6) Some ligands are shown in the figures but don't appear to be discussed in the text (at least that I can find), such as SR11237.

We thank the reviewer for pointing this out. We used a panel of 14 commercially available RXR ligands with different pharmacological properties to probe Nur77-RXRγ function, as in our previous Nurr1-RXRα study. In the text, we emphasized ligands that were most informative for the mechanistic conclusions, rather than discussing every compound individually. SR11237, for example, behaved similarly to the broader group of RXR agonists and was therefore shown as part of the full ligand panel but not specifically highlighted in the text. We will clarify this in the revised manuscript.

(7) There is a sentence in the discussion that says "these observations implicate that although RXRg LBD provides the protein-protein interaction interface to bind Nur77...." the authors did not show enough data to support this claim. It should be bolstered.

We thank the reviewer for this point. We agree that this statement was stronger than was warranted by the data presented. Our intent was not to claim that the present study definitively establishes the RXRγ LBD as the sole or fully defined protein-protein interaction interface for Nur77 binding. Rather, based on the domain truncation data together with our prior Nurr1-RXRα study, we intended this statement as a working interpretation consistent with an LBD-centered mechanism. In our revised manuscript, we have softened this language to avoid overstating the conclusion and clarified that the current data support, but do not definitively prove, a role for the RXRγ LBD in mediating functionally relevant interaction with Nur77.

**Reviewer #2 (Recommendations for the authors):**
Even though this study is not able to make definitive claims about the mechanism(s) of activation of Nur77 in the Nur77-RXRg complex, the work presented here is rigorous and solidly interpreted. Identifying differences between Nurr1 and Nur77 regulation is important, and the work here shows that selective agonists are essential for supporting the non-canonical mechanism they identified before. Although they address potential implications of NTD regulation in the discussion, it feels like a lot of insight into Nur77 regulation is being missed. However, it is clear that addressing this experimentally would require substantially more work. I don't have any specific recommendations. Given current limitations on funding, I think it's fine to focus on the work completed with the acceptance that it likely limits the impact of the work on the field.

We thank the reviewer for this thoughtful and balanced assessment of our work. The goal of this manuscript was to test whether the LBD heterodimer dissociation mechanism that we previously reported for Nurr1-RXRα may represent a conserved feature of NR4A-RXR heterodimers by extending these studies to Nur77-RXRγ. We agree that understanding the role of the RXRγ NTD in Nur77-RXRγ regulation is important and potentially highly informative. At the same time, resolving that question experimentally would require a distinct and more extensive set of studies beyond the scope of the present work. We have therefore chosen to focus this manuscript on the completed LBD-centered studies, while acknowledging that this narrower scope may limit the broader impact of the work.

Minor points:(1) Without page and line numbers, it is not easy to point out specific text. On the bottom of page 6 of the document, there are two references to Figure 3a, and the arrows that help illustrate RXRg LBD-dependent CSPs; the second figure callout should describe the blue arrow, I believe.

Thank you, we made this change.

(2) Bottom of page 8, "...revealed two compounds [that] standout..."

Thank you, we made this change.